# Health-related quality of life in a large cohort of patients with cardiac implantable electronic devices A registry-based study

Paolo Gatti[1]*, Carolin Nymark[2,3], Fredrik Gadler[1,3]

1 Division of Cardiology, Department of Medicine, Karolinska Institutet, Stockholm, Sweden, 2 Division of Nursing, Department of Neurobiology, Care Sciences and Society, Karolinska Institutet, Stockholm, Sweden, 3 Heart and Vascular Center, Karolinska University Hospital, Stockholm, Sweden

* paolo.gatti@ki.se

**Data Availability Statement:** The data utilized in this study are subject to ethical restrictions and are provided by the Swedish ICD and Pacemaker Registry. Access to these data is available upon

## Abstract

### Aim

The association of cardiac implantable electronic devices (CIED), namely pacemaker (PM), implantable cardioverter-defibrillator (ICD) and cardiac resynchronization therapy with (CRT-D) or without defibrillator (CRT-P) with health-related quality of life (HRQoL) is lacking.

### Methods and results

Data from the Swedish Pacemaker and ICD Registry collected from January 2019 to February 2022 was used to analyze the responses to the European Quality of Life-5 Dimension questionnaire (EQ-5D) before and after one year of the CIED implant. Descriptive analysis was performed using Pearson's chi-square test, the analysis of variance ANOVA, the Kruskal-Wallis test and Wilcoxon signed-rank test when appropriate. A multivariable regression analysis was used to compare the EQ-5D index and EQ-VAS variation after 1 year. Of 1,479 who completed the EQ-5D, 80% had a PM, 10% an ICD, 5% a CRT-P and 6% a CRT-D. The median age was 77 years with females constituting 38% of the PM group and 17% of the ICD group. The EQ-VAS and the EQ-5D index significantly increased after one year from the PM and CRT-P implant (EQ-VAS +2.8, standard deviations (SD) 23 and +5.8, SD 24.9; EQ-5D index +0.019, SD 0.114 and +0.051, SD 0.125) while only the EQ-5D index increased after one year from the ICD implant (+0.002, SD 0.104). After adjusting for age, sex and HRQoL at baseline, the presence of defibrillator was associated with lower EQ-VAS (ICD EQ-VAS variation: -3.4, 95% confidence intervals (CI) -6.7; -0.1 and CRT-D EQ-VAS variation -4.8, 95% CI -8.8;-0.7) and EQ-5D index (ICD EQ-5D index variation: -0.018, 95% CI -0.035; -0.0003 and CRT-D EQ-5D index variation -0.025 95% CI 0.046;0.004) after one year compared to PM.

### Conclusion

These findings, showing the HRQoL associated with CIED, are important to support physicians' and pacemaker nurses' care after device implantation by embracing the patients' perspectives.

request, contingent upon the submission of ethical approval. For further information and to request access to the data, please visit the registry's official website at https://www.pacemakerregistret.se/icdpmr/start.do.

**Funding:** The author(s) received no specific funding for this work.

**Competing interests:** The authors have declared that no competing interests exist.

## Introduction

According to the Swedish Pacemaker and implantable cardiac defibrillator Registry (annual report accessible online http://www.pacemakerregistret.se) annually more than 10,000 pacemakers (PM), implantable cardioverter defibrillator (ICD) or cardiac resynchronization therapy device (CRT) are implanted. Despite the large use of such therapies and their survival benefit, there is a substantial lack of research focused on their impact on health-related quality of life (HRQoL) [1]. HRQoL allows the measurement of the health status through patients' experience and not only with biological parameters. Quantifying a subjective perception in a measurable quantity is challenging and has its limitations. Several disease-specific and non-disease-specific patient-reported outcome measures (PROM) have been developed. Among non-disease-specific PROM the EuroQol 5-Dimension (EQ-5D) and the Short-Form Health Survey (SF-36) are the most widely used. Regarding cardiac implantable electronic device (CIED) there are few studies evaluating changes in HRQoL before and after the device implantation. The majority of studies evaluating HRQoL in patient implanted with a PM are based on the effect of different pacing modality [2–6]. Data on ICD and quality of life are more robust and did not show a relevant impact on HRQoL [7–10]. CRT devices instead had QoL improvement as a key outcome in randomized controlled trials as MIRACLE, REVERSE and lastly the BUDAPEST trial [11–13]. Several factors of medical treatment may impact the patient's perception. The level of information received about the device and the belief in the device's capacity to cure the disease underneath, especially if a new technology is considered, greatly impacts the HRQoL [14]. The level of information and amount of care varies in randomized clinical trials compared to registries and real-world scenarios [15]. Overestimating the efficacy of cardiac devices may also reduce patient compliance with other therapies and health-positive preventive behaviours [16]. Furthermore, sex and age may have different influences on HRQoL after device implantation [17]. The purpose of our registry based cohort study, therefore, is to describe the variation of HRQoL after one year from the CIED implantation helping to better understands and characterize the HRQoL domain status and variation of patients implanted with a CIED in a real-life setting.

## Methods

The study was conducted under the approval of the National Ethical Committee number: 2022-02429-01.

Unidentified data were accessed the 23$^{rd}$ January 2023 from the Swedish Pacemaker and ICD national quality registry.

Ethics committee of each participating hospital in the national quality registry approved data entry and collection. Individual patient consent was not required, but patients were informed of entry into national registries and allowed to opt out.

### Study population and registry data

All the patients implanted with a cardiac device are registered in the Swedish Pacemaker and ICD Registry and are offered to complete the EuroQol 5-Dimension 3-Level (EQ-5D-3L) questionnaire and the EQ visual analogue scale (EQ-VAS) [18] at the time of PM, ICD or CRT implantation and after 1 year with an online form or at the scheduled follow-up visit. Eligible patients were patients who completed the baseline and follow-up questionnaire from January 2019 to December 2021.

Independent variables available were: age expressed in years, sex, reported bradycardic indication for PM implantation (complete atrioventricular block, second-grade atrioventricular block, other atrioventricular blocks, sick sinus syndrome) and primary or secondary prevention indication for ICD implantation, pacing modality (only atrial pacing AAI, only

ventricular pacing mode VVI, bicameral pacing DDD) and reported prevalent symptomatology at the implant (no symptoms, dyspnea/asthenia, palpitations, syncope, dizziness).

### EQ-5D-3L and EQ-VAS

The EQ-5D-3L consists of 2 parts. Part 1 where the patients score 5 dimensions related to health aspects: mobility, self-care, usual activities, pain or discomfort, and anxiety or depression. Each dimension can be labelled by the patients in grades of severity as 1: no problems, 2: some problems, 3: extreme problems. Part 2 where the patients score their HRQoL on the EQ visual analogue scale (EQ-VAS) from 0 to 100 (full health). The Swedish value set for EQ-5D-3L questionnaire, which allows summarizing the HRQoL in part 1, attaching weights to each level of each dimension, was used to determine the global "EQ-5D index" from 0 to 1 (full health) [1]. If one or more of the dimension's levels were not reported, the patient was excluded from the analysis.

### Statistical analysis

In our retrospective paired cohort registry study, categorical variables were presented as absolute numbers (%) and continuous variables as mean [standard deviation (SD)] if normally distributed or medians [interquartile ranges (IQR)] if not normally distributed. Descriptive data analysis was performed using Pearson's chi-square test for comparing categorical variables across groups. The analysis of variance ANOVA and Kruskal-Wallis test was used for comparing means and medians of continuous variables across multiple groups when normally distributed and similar variance assumptions were met or not respectively. Wilcoxon signed-rank test was used to compare paired measurements not normally distributed. A multivariable tobit regression analysis was used to compare EQ-5D index and EQ-VAS variation after 1 year from the implant adjusting for age, sex and HRQoL at baseline.

All analyses were performed using Stata/IC version 16.1. The level of significance was set to 5%, two-sided.

## Results

During 2019, 9629 patients were implanted with a cardiac device in Sweden. Of these, 1816 patients (5.3%) completed at least the EQ-5D-3L or the EQ-VAS at baseline or at follow-up, and 1479 (81%) patients completed the questionnaire both at baseline and after one year of follow-up. Of those, 1177 (79.6%) were implanted with PM, 148 (10.0%) with ICD, 67 (4.5%) with CRT without defibrillator function (CRT-P) and 87 (5.9%) with CRT with defibrillator function (CRT-D) and were included in the analysis (Fig 1).

Patient characteristics are shown in Table 1. The median baseline EQ-VAS and EQ-5D index were 75 (IQR 60–85) and 0.9028 (IQR 0.8016–0.9694) respectively without significant difference between the devices group Table 1.

A separate table with detailed results for each single 5 dimensions by device type is reported in the supplement material with the representation of percentage variation after 1 year S1 Table and S1 Fig. The dimensions exploring pain and discomfort and anxiety and depression had a larger improvement after 1 year in all devices groups. Mobility and usual activity had a consistent improvement in the CRT-P group.

The global EQ-VAS and the EQ-5D index distributions are reported in S2 Fig.

According to device type, the EQ-VAS significantly improved after 1 year in the PM group (+2.8 SD 23). The EQ-5D index significantly improved after 1 year in the PM and CRT-P groups (+0.019 SD 0.114; +0.05 SD 0.125 respectively) (Fig 2) detailed values at baseline and after 1 year are reported in Table 1.

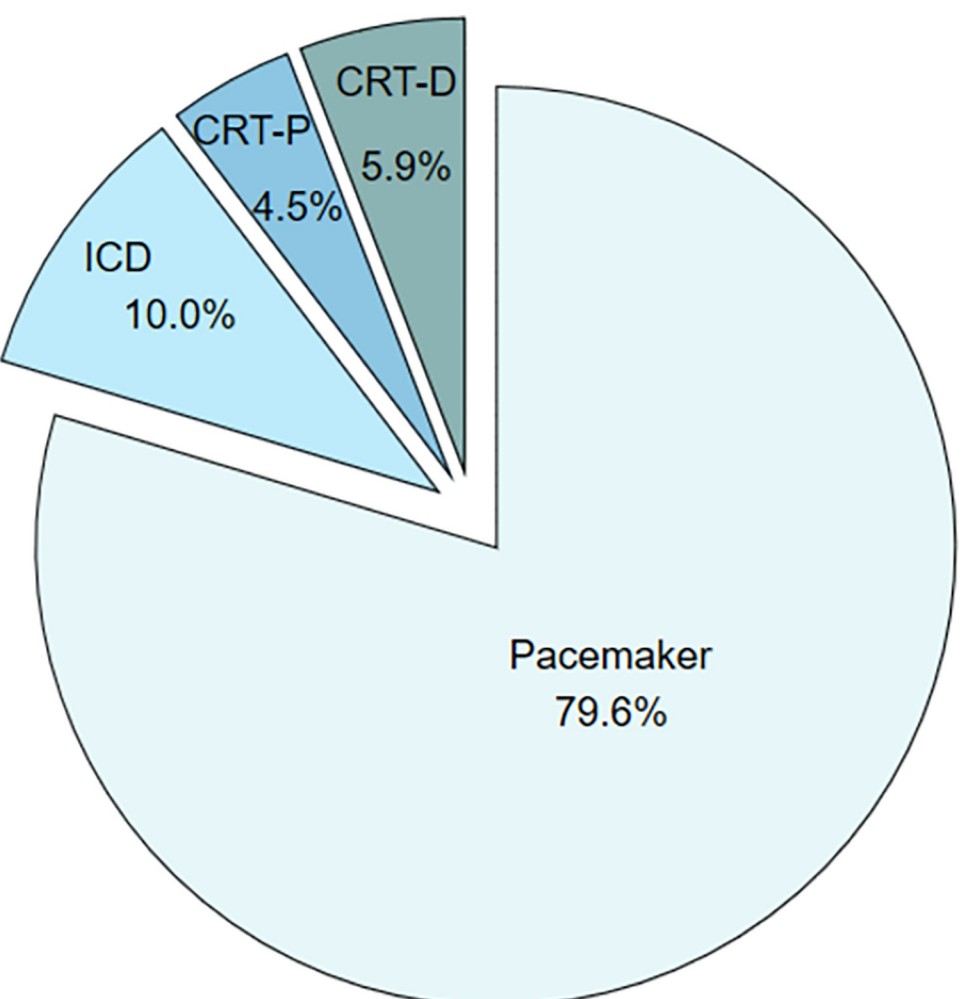

**Fig 1. Type of device prevalence.** CRT-D = cardiac resynchronization therapy with defibrillator function, CRT-P = cardiac resynchronization therapy without defibrillator function, EQ-5D-VAS = EuroQol 5-Dimension visual analogue scale, ICD = implantable cardioverter defibrillator, PM = pacemaker.

The symptoms reported as contributing to the device indication were distributed as follows: 433 (29%) dyspnea or asthenia, 178 (7%) palpitations, 455 (31%) syncope, 309 (21%) dizziness, and 178 (12%) asymptomatic. According to these symptoms, the EQ-VAS significantly improved in patients with dyspnea or asthenia, palpitation, syncope or dizziness but not in asymptomatic patients while the EQ-5D index significantly improved in all groups (Table 2 and Fig 3).

Variations of the EQ-VAS and the EQ-5D index by bradycardic indication, in patients implanted with PM, and by type of arrhythmic prevention, in patients implanted with ICD or CRT-D, are reported in the supplement material (S3 and S4 Figs). Patients implanted with a pacemaker had a significant improvement of EQ-VAS and EQ-5D index after one year if the principal reported bradycardic indication was sick sinus syndrome (SSS) or atrioventricular block of third-degree (AV-block III°) while no differences were detected for first-degree atrioventricular block (AV-block I°), or missing bradycardic indication. Patients with second-degree atrioventricular block (AV-block II°) as bradycardic indication had a significant improvement in the EQ-5D index but not in the EQ-VAS.

**Table 1. Baseline characteristics by type of device.**

| | Total | PM | ICD | CRT-P | CRT-D | p-value |
|---|---|---|---|---|---|---|
| | N = 1,479 | N = 1,177 (79%) | N = 148 (10%) | N = 67 (5%) | N = 87 (6%) | |
| **Male** | 974 (66%) | 735 (62%) | 123 (83%) | 49 (73%) | 67 (77%) | <0.001 |
| **Age** | 77 (71; 82) | 78 (73; 83) | 66 (56; 73) | 79 (75; 83) | 70 (62; 76) | <0.001 |
| **pacing modality** | | | | | | <0.001 |
| **AAIR** | 2 (0%) | 2 (0%) | 0 (0%) | 0 (0%) | 0 (0%) | |
| **DDDR** | 1,267 (86%) | 1,009 (86%) | 104 (70%) | 67 (100%) | 87 (100%) | |
| **VVIR** | 210 (14%) | 166 (14%) | 44 (30%) | 0 (0%) | 0 (0%) | |
| **EQ-VAS at baseline** | 75 (60; 85) | 75 (60; 85) | 75 (63; 88) | 70 (50; 80) | 75 (60; 85) | 0.09 |
| **EQ-VAS at follow-up** | 77 (65; 90) | 80 (65; 90) | 76 (65; 90) | 75 (60; 85) | 75 (60; 90) | 0.33 |
| **EQ-VAS variation** | 2.6 (22.7) | 2.8 (23.0) | 1.8 (20.8) | 5.8 (24.9) | -0.7 (19.6) | 0.32 |
| **EQ-5D index at baseline** | .9028 (.8016; .9694) | .9028 (.8016; .9694) | .9142 (.8131; .9694) | .8797 (.7671; .9694) | .9142 (.813; .9694) | 0.17 |
| **EQ-5D index at follow-up** | .9142 (.8131; .9694) | .9142 (.8131; .9694) | .9349 (.8233; .9694) | .9142 (.8357; .9694) | .9142 (.8336; .9694) | 0.78 |
| **EQ-5D index variation** | .0188 (.1142) | .0188 (.1139) | .0155 (.116) | .0512 (.1253) | -.0015 (.1019) | 0.04 |
| **Bradycardic indication** | | | | | | <0.001 |
| **AV delay** | 55 (4%) | 37 (3%) | 10 (7%) | 3 (4%) | 5 (6%) | |
| **AV-block II°** | 251 (17%) | 235 (20%) | 1 (1%) | 11 (16%) | 4 (5%) | |
| **AV-block III°** | 344 (23%) | 316 (27%) | 3 (2%) | 14 (21%) | 11 (13%) | |
| **SSS** | 443 (30%) | 425 (36%) | 10 (7%) | 3 (4%) | 5 (6%) | |
| **Missing** | 386 (26%) | 164 (14%) | 124 (84%) | 36 (54%) | 62 (71%) | |
| **Tachycardic indication** | | | | | | <0.001 |
| **primary prevention** | 159 (11%) | 0 (0%) | 78 (53%) | 0 (0%) | 81 (93%) | |
| **secondary prevention** | 76 (5%) | 0 (0%) | 70 (47%) | 0 (0%) | 6 (7%) | |
| **Missing** | 1,244 (84%) | 1,177 (100%) | 0 (0%) | 67 (100%) | 0 (0%) | |
| **Symptom** | | | | | | <0.001 |
| **Dyspnea/asthenia** | 433 (29%) | 264 (22%) | 36 (24%) | 56 (84%) | 77 (89%) | |
| **Asymptomatic** | 178 (12%) | 95 (8%) | 78 (53%) | 2 (3%) | 3 (3%) | |
| **Palpitations** | 104 (7%) | 88 (7%) | 12 (8%) | 1 (1%) | 3 (3%) | |
| **Syncope** | 455 (31%) | 425 (36%) | 22 (15%) | 4 (6%) | 4 (5%) | |
| **Dizziness** | 309 (21%) | 305 (26%) | 0 (0%) | 4 (6%) | 0 (0%) | |

AAIR = atrial pacing atrial sensing inhibited-response rate-adaptive, AV-block = atrioventricular block, CRT-D = cardiac resynchronization therapy with defibrillator function, CRT-P = cardiac resynchronization therapy without defibrillator function, DDDR = dual-chamber atrioventricular pacing atrioventricular sensing rate-adaptive, EQ-5D = EuroQol 5-Dimension, EQ-VAS = EuroQol visual analogue scale, ICD = implantable cardioverter defibrillator, PM = pacemaker, SSS = sick sinus syndrome, VVIR = ventricular pacing ventricular sensing inhibited-response rate-adaptive.

Patients with ICD or CRT-D implanted as secondary prevention had a significant improvement in the EQ-5D index after one year.

After adjusting for age and sex and HRQoL and symptoms at baseline, the ICD and CRT-D group had a significantly lower EQ-5D index (-0.0215, 95% confidence intervals (CI) -0.0105; -0.0024 and -0.0257, 95% CI -0.0482;-0.0032) after 1 year compared to the pacemaker group.

Age but not sex or symptoms at the implant was associated with worse EQ-VAS and EQ-5D index after device implantation (-2.5, 95% CI -3.4;-1.5 and -0.0153 95% CI -0.0202;-0.0103 respectively) Fig 4.

## Discussion

Our study aims to describe the HRQoL status of patients receiving cardiac implantable electronic device (CIED) in a large national cohort using the HRQoL EQ-5D-3L questionnaire at

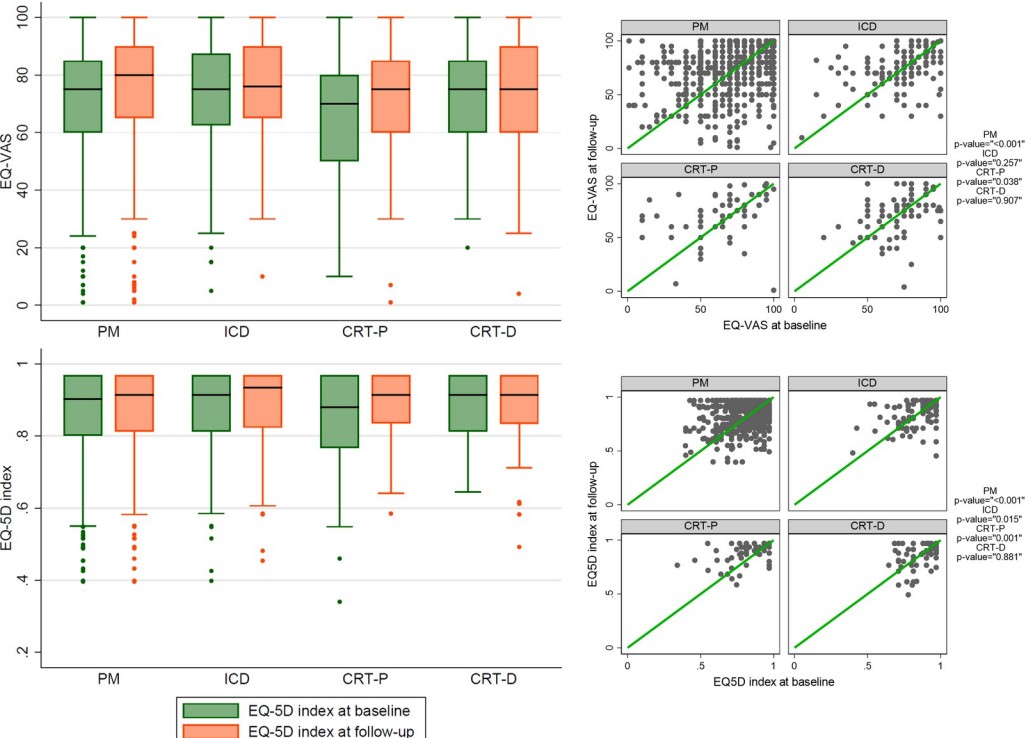

**Fig 2. EQ-5D visual analogue scale and EQ-5D index variation after 1 year by type of device.** CRT-D = cardiac resynchronization therapy with defibrillator function, CRT-P = cardiac resynchronization therapy without defibrillator function, EQ-5D index = EuroQol 5-Dimension index, EQ-VAS = EuroQol visual analogue scale, ICD = implantable cardioverter defibrillator, PM = pacemaker.

the implant and at one-year follow-up. CIED efficacy and effectiveness have a large core of evidence used by the recent European and American Guidelines on cardiac pacing to provide clinical indications [19, 20]. While the impact of CIED on HRQoL had been studied in clinical trials especially for ICD and CRT or for pacing modality (CRT and single vs dual chamber pacing) in PM [3, 11, 21–24], and since the current state of pharmacological treatment has changed the current state of the art in a real-world setting is lacking.

The first interesting finding of our study was the high EQ-5D index and EQ-VAS value at baseline. Our cohort of patients mainly implanted with pacemaker had an HRQoL at baseline comparable with the general Swedish population [25]. In the study of Inácio N.A. et al. on elderly patients with pacemakers the median EQ5D index was lower but similar EQ-VAS was reported [26]. Nevertheless, a direct comparison with other studies on CIED is difficult due to the variety of population groups and questionnaires utilised (SF36, MLHF, MacNew scale) [27–29]. Other than patient characteristics and comorbidities that have a heavy impact on HRQoL, environmental factors may play a role. We think that the online subministration of the questionnaire allowing the completion at home, in a less stressful environment may have also impacted our results.

In line with results from previous studies and meta-analyses, [2, 28, 30] our cohort showed an overall improvement in HRQoL at 1 year from the CIED implantation. The improvement of HRQoL was significant in patients implanted with PM or CRT-P but not for ICD and CRT-D. The defibrillation function by its nature differs from pacing, as it is comparable to insurance. While it is adopted based on the patient's risk of sudden death, the hope is that it

**Table 2. Characteristics by symptoms at the implant.**

| | Total | Dyspnea/asthenia | Asymptomatic | Palpitations | Syncope | Dizziness | p-value |
|---|---|---|---|---|---|---|---|
| | N = 1,479 | N = 433 (29%) | N = 178 (12%) | N = 104 (7%) | N = 455 (31%) | N = 309 (21%) | |
| **Sex** | 974 (66%) | 298 (69%) | 130 (73%) | 57 (55%) | 296 (65%) | 193 (62%) | 0.011 |
| **Age** | 77 (71; 82) | 76 (70; 80) | 73 (61; 80) | 77 (71; 83) | 78 (72; 83) | 78 (74; 83) | <0.001 |
| **Device type** | | | | | | | <0.001 |
| **PM** | 1,177 (80%) | 264 (61%) | 95 (53%) | 88 (85%) | 425 (93%) | 305 (99%) | |
| **ICD** | 148 (10%) | 36 (8%) | 78 (44%) | 12 (12%) | 22 (5%) | 0 (0%) | |
| **CRT-P** | 67 (5%) | 56 (13%) | 2 (1%) | 1 (1%) | 4 (1%) | 4 (1%) | |
| **CRT-D** | 87 (6%) | 77 (18%) | 3 (2%) | 3 (3%) | 4 (1%) | 0 (0%) | |
| **pacing modality** | | | | | | | 0.082 |
| **AAIR** | 2 (0%) | 0 (0%) | 0 (0%) | 1 (1%) | 0 (0%) | 1 (0%) | |
| **DDDR** | 1,267 (86%) | 377 (87%) | 143 (80%) | 89 (86%) | 398 (87%) | 260 (84%) | |
| **VVIR** | 210 (14%) | 56 (13%) | 35 (20%) | 14 (13%) | 57 (13%) | 48 (16%) | |
| **EQ-VAS at baseline** | 75 (60; 85) | 75 (55; 85) | 75 (60; 90) | 75 (57; 85) | 75 (56; 90) | 75 (60; 85) | 0.33 |
| **EQ-VAS at follow-up** | 77 (65; 90) | 75 (60; 85) | 80 (67; 90) | 80 (65; 90) | 77 (65; 90) | 75 (65; 90) | 0.36 |
| **EQ-VAS variation** | 2.6 (22.7) | 2.3 (22.2) | 2.0 (22.6) | 4.4 (19.9) | 2.3 (23.6) | 3.3 (23.0) | 0.87 |
| **EQ-5D index at baseline** | .9028 (.8016; .9694) | .9028 (.7805; .9694) | .9142 (.8336; .9694) | .8797 (.7784; .9694) | .9028 (.7855; .9694) | .8797 (.8131; .9349) | 0.25 |
| **EQ-5D index at follow-up** | .9142 (.8131; .9694) | .9142 (.8336; .9694) | .9349 (.8407; .9694) | .92455 (.8336; .9694) | .9142 (.8016; .9694) | .9142 (.82; .9694) | 0.38 |
| **EQ-5D index variation** | .0188 (.1142) | .0215 (.1128) | .0198 (.1213) | .0283 (.1049) | .0118 (.1105) | .0213 (.1202) | 0.59 |

CRT-D = cardiac resynchronization therapy with defibrillator function, CRT-P = cardiac resynchronization therapy without defibrillator function, EQ-5D

index = EuroQol 5-Dimension index, EQ-VAS = EuroQol visual analogue scale, ICD = implantable cardioverter defibrillator, PM = pacemaker

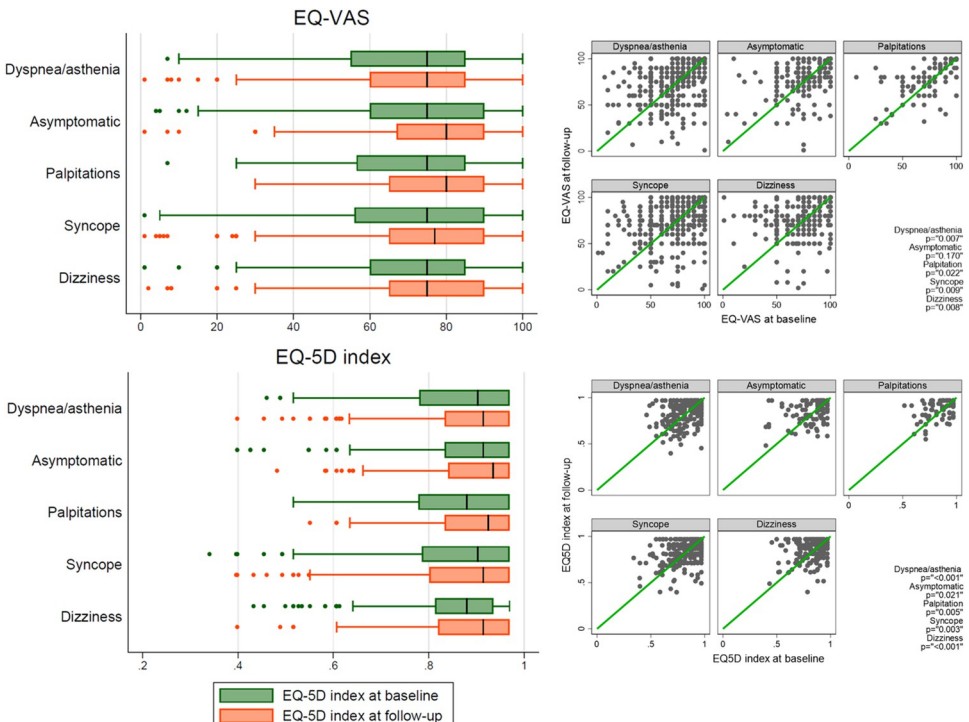

**Fig 3. EQ-5D visual analogue scale and EQ-5D index variation after 1 year by symptoms.** EQ-5D index = EuroQol
5-Dimension index, EQ-VAS = EuroQol visual analogue scale.

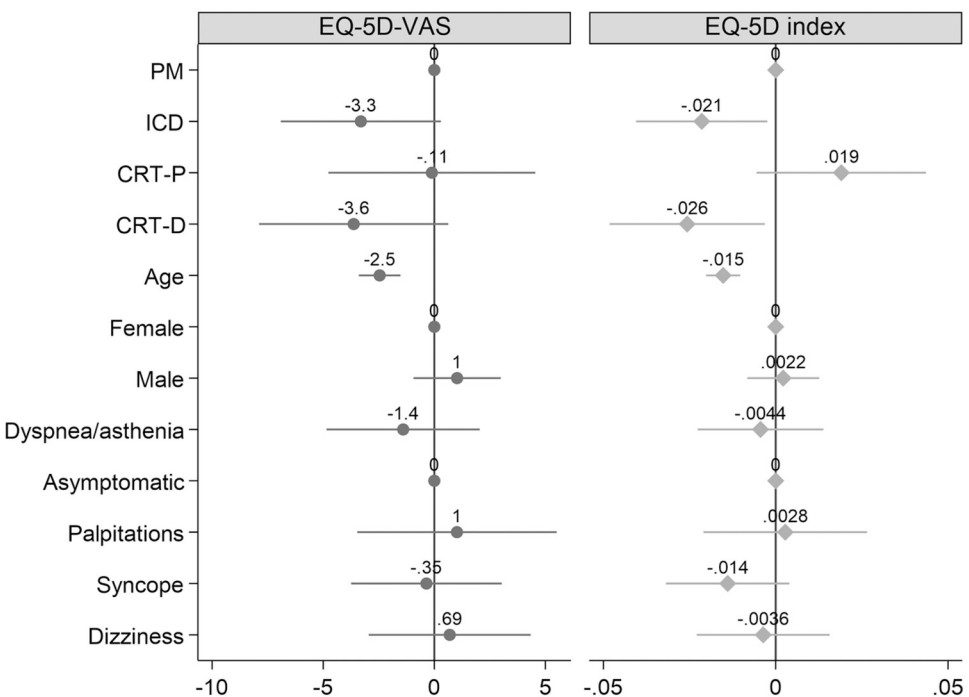

**Fig 4. Predictors of EQ-5D visual analogue scale and EQ-5D index variation after 1 year.** CRT-D = cardiac resynchronization therapy with defibrillator function, CRT-P = cardiac resynchronization therapy without defibrillator function, EQ-5D-VAS = EuroQol 5-Dimension visual analogue scale, ICD = implantable cardioverter defibrillator, PM = pacemaker.

will not need to be used. For this reason, the lack of tangible effects on symptoms in the absence of a precipitating event that triggers its activation, along with the fear of inappropriate shocks, a higher rate of complication and restrictions on driving license and specific activities may have counteracted other beneficial effects perceived by the patients [10, 21, 31, 32].

Interestingly, the domains with greater improvement were pain or discomfort and anxiety or depression. These results show the important weight of psychological aspects in patients' perspectives. Patients with CRT-P showed a large improvement in mobility and usual activities that may reflect the device's impact on functional capacity as well as a different health status perception in the population implanted with CRT-P that are generally older with more comorbidities [23].

Despite the limitation of subgroup analysis, we described available subgroups in our population such as different symptoms and indications at the CIED implant, to generate hypotheses and further advance the knowledge of patients' reported outcomes.

Symptoms impact HRQoL and from the patient's perspective, may be more relevant than outcomes such as mortality or hospitalization. In our cohort patients with symptoms correlated to the device indication, in contrast to asymptomatic patients, had a significant improvement in HRQoL. After adjusting for age, sex, and HRQoL at baseline, no significant differences were detected in HRQoL variation for any type of symptoms compared to asymptomatic patients. Hence the symptoms leading to PM implantation may affect patients' HRQoL but further dedicated studies have to be done. The Atrioventricular (AV) -block III° or Sick sinus syndrome (SSS) indication to PM showed a significant improvement on HRQoL but not AV-block I° or the absence of reported bradycardic indication. This finding may suggest the possibility that strong and clear indications may not only have a greater impact on biological outcomes but also on patients' perspectives.

Secondary prevention for ICD and CRT-D has a stronger indication but differences between primary and secondary prevention on HRQoL are controversial [33, 34]. Our study showed a significant improvement in HRQoL that was not replicated in the primary prevention indication. These findings may raise the hypothesis, as suggested in the European Heart Rhythm Association survey on long-term HRQoL and acceptance of implantable ICD [32], that patients who experience major arrhythmias had a larger psychological benefit from the security provided by the defibrillator who may overcome the fear of complications.

After adjusting for age, sex, symptoms and the baseline HRQoL the devices with defibrillator ICD and CRT-D had a lower HRQoL compared to patients with PM. Although the effect of defibrillators on HRQoL may not be negligible, these results are mainly explained by the different cardiac conditions and the different device choices that were not accounted for in our study.

In the previous European survey regarding living with CIED, older patients perceived more frequently an improvement of HRQoL connected to the device implantation independently from the type of device (PM vs ICD/CRT) [1, 35]. In our study, older age was associated with reduced HRQoL after 1 year in patients implanted with CIED. This can be explained by several factors including the different study designs (an instant survey versus a longitudinal measurement). Our study directly measured the HRQoL with the EQ-5D-3L questionnaire without specifically addressing the acceptance of the device. Younger patients may have had higher concerns about device complications and daily activity limitation but a better overall HRQoL compared to baseline.

## Limitations

Our analysis is affected by several limitations. Important information like medical history and socio-economic status were not available in our analysis. Nevertheless, our studies aimed to provide a picture of the current situation in a large population without deriving causal implications that only randomized clinical trials can achieve. Therefore, the lack of randomization and the impossibility of adjusting for several factors that impact the HRQoL prevent causal inference and reported implications should be considered as speculative and hypothesis generator. Our results are not generalizable to other countries since all the information came from the Swedish national pacemaker and ICD registry and the adapted to Swedish population EQ-5D-3L value set was used to determine the EQ index. Patients' characteristics and device indications such as the percentage of secondary and primary prevention for defibrillators reflect the Swedish health care system and may not apply to other countries.

A selection bias due to patients' acceptance in completing the EQ-5D-3L questionnaire cannot be ruled out and may affect our results. The baseline EQ-5D-3L questionnaire was assessed at the time of the device implantation procedure which potentially increased the health problem perception leading to lower HRQoL results compared to follow-up. The exclusion of patients who did not complete the follow-up questionnaire limited the generalizability since the chance in HRQoL was not based on the entire population. However, 81% of the screened cohort completed the questionnaire both at baseline and at follow-up.

## Conclusion

The treatment with CIED is one of the purest curative interventions. Nevertheless, accounting for patients' reported outcomes and HRQoL is fundamental. The picture provided by this large real-world cohort shows no worsening of HRQoL after CIED implant and strengthens the importance of including the HRQoL in the decision to implement a device, and in further study designs.

## Supporting information

**S1 Fig. Detailed dimension variation at 1 year by device type.** CRT-D = Cardiac resynchronization therapy with defibrillator function, CRT-P = Cardiac resynchronization therapy without defibrillator function, ICD = implantable cardioverter defibrillator, PM = pacemaker.
(TIF)

**S2 Fig. EQ visual analogue scale and EQ-5D index distributions at baseline and after 1 year.** EQ-5D = EuroQol 5-Dimension, EQ-VAS = EuroQol visual analogue scale.
(TIF)

**S3 Fig. EQ-5D visual analogue (EQ-5D-VAS) scale and EQ-5D index variation after 1 year in patients with pacemakers by bradycardic indication at the implantation.**
AV = atrioventricular, EQ-5D index = EuroQol 5-Dimension index, EQ-VAS = EuroQol visual analogue scale, SSS = sick sinus syndrome.
(TIF)

**S4 Fig. EQ-5D visual analogue scale (EQ-5D-VAS) and EQ-5D index variation after 1 year in patients with implantable cardioverter defibrillator or cardiac resynchronization therapy with defibrillator function by type of arrhythmic prevention.** EQ-5D index = EuroQol 5-Dimension index, EQ-VAS = EuroQol visual analogue scale.
(TIF)

**S1 Table. EQ-5D dimensions result by device type at baseline and after 1 year.** n = number, CRTD = Cardiac resynchronization therapy with defibrillator function, CRT-P = Cardiac resynchronization therapy without defibrillator function, ICD = implantable cardioverter defibrillator, PM = pacemacker.
(DOCX)

## Author Contributions

**Conceptualization:** Carolin Nymark, Fredrik Gadler.

**Data curation:** Paolo Gatti, Fredrik Gadler.

**Formal analysis:** Paolo Gatti.

**Methodology:** Fredrik Gadler.

**Resources:** Fredrik Gadler.

**Supervision:** Fredrik Gadler.

**Writing – original draft:** Paolo Gatti.

**Writing – review & editing:** Paolo Gatti, Carolin Nymark, Fredrik Gadler.

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
