## [Decision Letter · Decision Letter 0]

16 Jul 2024

PONE-D-24-18282Health-related quality of life in in a large cohort of patients with cardiac implantable electronic devices – a registry based studyPLOS ONE

Dear Dr. Gatti,

Thank you for submitting your manuscript to PLOS ONE. After careful consideration, we feel that it has merit but does not fully meet PLOS ONE’s publication criteria as it currently stands. Therefore, we invite you to submit a revised version of the manuscript that addresses the points raised during the review process.

**ACADEMIC EDITOR: **

While the manuscript exhibits intriguing potential, it requires significant revisions and further refinement.<o:p></o:p>

Although the inherent interest in the subject matter is acknowledged, the reviewers have raised crucial concerns that must be properly addressed.

We look forward to receiving your revised manuscript.

Kind regards,

Marcelo Arruda Nakazone, M.D., Ph.D.

Academic Editor

PLOS ONE

Additional Editor Comments (if provided):

Reviewers' comments:

Reviewer's Responses to Questions

**Comments to the Author**

1. Is the manuscript technically sound, and do the data support the conclusions?

Reviewer #1: No

Reviewer #2: Yes

2. Has the statistical analysis been performed appropriately and rigorously? 

Reviewer #1: I Don't Know

Reviewer #2: Yes

3. Have the authors made all data underlying the findings in their manuscript fully available?

Reviewer #1: No

Reviewer #2: No

4. Is the manuscript presented in an intelligible fashion and written in standard English?

Reviewer #1: Yes

Reviewer #2: Yes

5. Review Comments to the Author

Reviewer #1: Thank you for the opportunity to revise this manuscript. The authors evaluated the association of cardiac implantable electronic devices with health-related quality of life (HRQoL). While the article is grammatically sound, it requires a more comprehensive explanation of the research methodology.

MAJOR COMMENTS

Abstract

1. Please describe the statistical analyses performed in the methods section.

2. What is the reference group in these analyses?

3. In this sentence in lines 14-15 "the presence of defibrillator was associated with lower EQ-VAS (-3.4, 95% confidence intervals (CI) -6.7; -0.1 and -4.8, 95% CI -8.8;-0.7)", what do these coefficients represent?

4. What is the study design?

5. Please indicate that HRQoL measurements were conducted twice.

Introduction

The authors ought to incorporate a literature review detailing what is already known about the topic, identifying the knowledge gaps in existing literature, and explaining how this study addresses those gaps.

Methods:

1. Please provide a comprehensive description of the independent variables in this analysis

2. Statistical Analysis - Line 10: What type of regression was used? This question is particularly important because the outcome variables are bounded (EQ-VAS ranges from 1-100)

3. What is the study design?

MINOR COMMENTS

Line 4: Please remove the comma after "(CRT-P)" and change" [on] health-related quality of life (HRQoL)" to "[with] health-related quality of life (HRQoL) "

Line 8: Please replace "February 2022 to analyze" with "February 2022 was used to analyze"

Lines 9-10: Please consider rephrasing this sentence to make the meaning of "prevalence of females" clearer. Does it mean the proportion of females? If so, the sentence could be revised as follows: "The median age was 77 years with females constituting 38% of the PM group and 17% of the ICD group.

Reviewer #2: This study asks and addresses an important question – what are the effects of cardiac implantable electronic devices (CIED) on the well-being of patients? Importantly, this assessment of well-being is performed by the patients themselves. This paper is well written and presents a series of well-explained and sensible analyses. Understanding the impact of various variables on post-implantation well-being is clearly important, and this paper makes an important contribution to this area of inquiry.

It is interesting (and surprising to me) that the baseline HRQoL values seem so high? For example, the median baseline EQ-5D index is greater than 0.9. Isn’t this a surprisingly high number for patients about to undergo surgery? I think it would be useful for the authors to discuss this and comment on the range of values observed in other relevant groups e.g. healthy population.

Minor edits -

Typo in title! “in in”

Abstract –

Change “one year the” to “one year of the”

6. PLOS authors have the option to publish the peer review history of their article (what does this mean?). If published, this will include your full peer review and any attached files.

Reviewer #1: No

Reviewer #2: **Yes: **Akshay Bareja

---

## [Author Response · Author response to Decision Letter 0]

19 Aug 2024

We are extremely grateful for the efforts of the reviewer on our resubmission, which have permitted a considerable improvement of the manuscript. A point-to-point response to each issue raised is reported in the "response to reviewer" and changes are identified by red font in the “Revised Manuscript with Track Changes”.

---

## [Decision Letter · Decision Letter 1]

30 Sep 2024

PONE-D-24-18282R1Health-related quality of life in a large cohort of patients with cardiac implantable electronic devices A registry-based studyPLOS ONE

Dear Dr. Gatti,

Thank you for submitting your manuscript to PLOS ONE. After careful consideration, we feel that it has merit but does not fully meet PLOS ONE’s publication criteria as it currently stands. Therefore, we invite you to submit a revised version of the manuscript that addresses the points raised during the review process.

**ACADEMIC EDITOR: **

Thank you for addressing the previous comments. The manuscript presents a subject of potential interest. However, one of the reviewers has raised questions about specific aspects that need to be properly addressed. Please ensure that these points are considered in your revision to enhance the manuscript's clarity and depth.

We look forward to receiving your revised manuscript.

Kind regards,

Marcelo Arruda Nakazone, M.D., Ph.D.

Academic Editor

PLOS ONE

Journal Requirements:

Reviewers' comments:

Reviewer's Responses to Questions

**Comments to the Author**

1. If the authors have adequately addressed your comments raised in a previous round of review and you feel that this manuscript is now acceptable for publication, you may indicate that here to bypass the “Comments to the Author” section, enter your conflict of interest statement in the “Confidential to Editor” section, and submit your "Accept" recommendation.

Reviewer #2: All comments have been addressed

Reviewer #3: All comments have been addressed

Reviewer #4: All comments have been addressed

2. Is the manuscript technically sound, and do the data support the conclusions?

Reviewer #2: Yes

Reviewer #3: Yes

Reviewer #4: Yes

3. Has the statistical analysis been performed appropriately and rigorously? 

Reviewer #2: Yes

Reviewer #3: Yes

Reviewer #4: Yes

4. Have the authors made all data underlying the findings in their manuscript fully available?

Reviewer #2: No

Reviewer #3: Yes

Reviewer #4: Yes

5. Is the manuscript presented in an intelligible fashion and written in standard English?

Reviewer #2: Yes

Reviewer #3: Yes

Reviewer #4: Yes

6. Review Comments to the Author

Reviewer #2: (No Response)

Reviewer #3: Authors are undertaking an interesting task of comparing quality of life changes associated with CIED implantation.

Needless to say, study is hindered by only analyzing answers of motivated patients and we should keep that in mind.

I have questions requiring clarification

“since its preventive nature instead of physical symptomatology treatment, may no impact patients' reported outcomes… Furthermore, the fear of inappropriate shocks, a higher

rate of complication and restrictions on driving license and specific activities may have

counteracted other beneficial effects perceived by the patients”

Very confusing statement to me since I see only grouped analysis of ICD implants without break down for primary and secondary prevention and I think these should that be analyzed separately

“Our study showed a significant improvement in HRQoL that was not replicated in the

primary prevention indication. These findings may show, as reported in the European Heart

Rhythm Association survey on long-term HRQoL and acceptance of implantable ICD (32), that

patients who experience major arrhythmias had a larger psychological benefit from the security

193 provided by the defibrillator who may overcome the fear of complication”

Again, I do not see the comparison. What worries me is that secondary prevention was an indication in nearly half of the ICD group

No worsening of symptoms of quality of life needs to be highlighted I consider that a major finding of this study. My view is that any improvement is a bonus for many patients, particularly those with heart failure.

Reviewer #4: Very interesting article and addresses a very important subject that is the quality of life of people with implantable electronic cardiac devices

7. PLOS authors have the option to publish the peer review history of their article (what does this mean?). If published, this will include your full peer review and any attached files.

Reviewer #2: No

Reviewer #3: No

Reviewer #4: No

---

## [Author Response · Author response to Decision Letter 1]

18 Oct 2024

Review Comments:

Reviewer 3:

1. “Since its preventive nature instead of physical symptomatology treatment, may no impact patients' reported outcomes… Furthermore, the fear of inappropriate shocks, a higher

rate of complication and restrictions on driving license and specific activities may have

counteracted other beneficial effects perceived by the patients”

Very confusing statement to me since I see only grouped analysis of ICD implants without break down for primary and secondary prevention and I think these should that be analyzed separately

REPLY

We thank the reviewer for his/her comments. We agree with the reviewer that the sentence as written was unclear; the term "preventive nature" is misleading. The intent was to emphasize the different nature of the defibrillation function compared to pacing. For this reason, we have modified the paragraph accordingly

See Discussion, page 10 line 165.

- The defibrillation function by its nature differs from pacing, as it is comparable to insurance. While it is adopted based on the patient's risk of sudden death, the hope is that it will not need to be used. For this reason, the lack of tangible effects on symptoms in the absence of a precipitating event that triggers its activation, along with the fear of inappropriate shocks, a higher rate of complication and restrictions on driving license and specific activities may have counteracted other beneficial effects perceived by the patients.

2. “Our study showed a significant improvement in HRQoL that was not replicated in the

primary prevention indication. These findings may show, as reported in the European Heart

Rhythm Association survey on long-term HRQoL and acceptance of implantable ICD (32), that

patients who experience major arrhythmias had a larger psychological benefit from the security provided by the defibrillator who may overcome the fear of complication”

Again, I do not see the comparison. What worries me is that secondary prevention was an indication in nearly half of the ICD group

REPLY

We thank the reviewer for his/her comments. Indeed, as suggested by the reviewer, a comparison of subgroups, such as between primary and secondary prevention, especially in an observational study does not allow for conclusions. Despite this, we considered it useful to present our results in the supplementary material with the intent of describing the population and generating hypotheses. 

As rightly pointed out, the percentage of patients with ICDs implanted for secondary prevention is high compared to other countries. For this reason, we have reduced the emphasis on subgroup results in our discussion and highlighted the limitations in the limitations section.

See Discussion, page 11 line 178, 189, 195

- Despite the limitation of subgroup analysis, we described available subgroups in our population such as different symptoms and indications at the CIED implant, to generate hypotheses and further advance the knowledge of patients' reported outcomes.

… This finding may suggest the possibility that strong and clear indications may not only have a greater impact on biological outcomes but also on patients’ perspectives.

… These findings may raise the hypothesis, as suggested in the European Heart Rhythm Association survey on long-term HRQoL and acceptance of implantable ICD, that patients who experience major arrhythmias had a larger psychological benefit from the security provided by the defibrillator who may overcome the fear of complications. 

See Limitation, page 13 line 2020

Patients’ characteristics and device indications such as the percentage of secondary and primary prevention for defibrillators reflect the Swedish health care system and may not apply to other countries.

3. No worsening of symptoms of quality of life needs to be highlighted I consider that a major finding of this study. My view is that any improvement is a bonus for many patients, particularly those with heart failure.

REPLY

We thank the reviewer for his/her comments. We agree that no worsening of quality of life in patients receiving a device is the most important result. 

Se Coclusion, Page 13 line 232

---

## [Decision Letter · Decision Letter 2]

20 Nov 2024

Health-related quality of life in a large cohort of patients with cardiac implantable electronic devices

A registry-based study

PONE-D-24-18282R2

Dear Dr. Gatti,

We’re pleased to inform you that your manuscript has been judged scientifically suitable for publication and will be formally accepted for publication once it meets all outstanding technical requirements.

Kind regards,

Marcelo Arruda Nakazone, M.D., Ph.D.

Academic Editor

PLOS ONE

Additional Editor Comments (optional):

Reviewers' comments:

Reviewer's Responses to Questions

**Comments to the Author**

1. If the authors have adequately addressed your comments raised in a previous round of review and you feel that this manuscript is now acceptable for publication, you may indicate that here to bypass the “Comments to the Author” section, enter your conflict of interest statement in the “Confidential to Editor” section, and submit your "Accept" recommendation.

Reviewer #3: All comments have been addressed

2. Is the manuscript technically sound, and do the data support the conclusions?

Reviewer #3: Yes

3. Has the statistical analysis been performed appropriately and rigorously? 

Reviewer #3: Yes

4. Have the authors made all data underlying the findings in their manuscript fully available?

Reviewer #3: Yes

5. Is the manuscript presented in an intelligible fashion and written in standard English?

Reviewer #3: Yes

6. Review Comments to the Author

Reviewer #3: acceptable in current format

7. PLOS authors have the option to publish the peer review history of their article (what does this mean?). If published, this will include your full peer review and any attached files.

Reviewer #3: No

---

## [Editor Report · Acceptance letter]

29 Nov 2024

PONE-D-24-18282R2 

PLOS ONE

Dear Dr. Gatti, 

I'm pleased to inform you that your manuscript has been deemed suitable for publication in PLOS ONE. Congratulations! Your manuscript is now being handed over to our production team.

Kind regards, 

on behalf of

Professor Marcelo Arruda Nakazone 

Academic Editor

PLOS ONE